

# Novel genetic variants in long non-coding RNA MEG3 are associated with the risk of asthma

Kuo-Liang Chiu[1,2], Wen-Shin Chang[3,4], Chia-Wen Tsai[3,4], Mei-Chin Mong[5], Te-Chun Hsia[4] and Da-Tian Bau[3,4,6]

[1] Division of Chest Medicine, Department of Internal Medicine, Taichung Tzu Chi Hospital, Taichung, Taiwan
[2] School of Post-Baccalaureate Chinese Medicine, Tzu Chi University, Hualien, Taiwan
[3] Graduate Institute of Biomedical Sciences, China Medical University, Taichung, Taiwan
[4] Terry Fox Cancer Research Laboratory, Department of Medical Research, China Medical University Hospital, Taichung, Taiwan
[5] Department of Food Nutrition and Health Biotechnology, Asia University, Taichung, Taiwan
[6] Department of Bioinformatics and Medical Engineering, Asia University, Taichung, Taiwan

Corresponding authors
Te-Chun Hsia,
derrick.hsia@msa.hinet.net
Da-Tian Bau, artbau2@gmail.com

## ABSTRACT

**Background.** Asthma is the most common chronic inflammatory airway disease worldwide. Asthma is a complex disease whose exact etiologic mechanisms remain elusive; however, it is increasingly evident that genetic factors play essential roles in the development of asthma. The purpose of this study is to identify novel genetic susceptibility loci for asthma in Taiwanese. We selected a well-studied long non-coding RNA (lncRNA), *MEG3*, which is involved in multiple cellular functions and whose expression has been associated with asthma. We hypothesize that genetic variants in *MEG3* may influence the risk of asthma.

**Methods.** We genotyped four single nucleotide polymorphisms (SNPs) in *MEG3*, rs7158663, rs3087918, rs11160608, and rs4081134, in 198 patients with asthma and 453 healthy controls and measured serum *MEG3* expression level in a subset of controls.

**Results.** The variant AG and AA genotypes of *MEG3* rs7158663 were significantly over-represented in the patients compared to the controls ($P = 0.0024$). In logistic regression analyses, compared with the wild-type GG genotype, the heterozygous variant genotype (AG) was associated with a 1.62-fold [95% confidence interval (CI) [1.18–2.32], $P = 0.0093$] increased risk and the homozygous variant genotype (AA) conferred a 2.68-fold (95% CI [1.52–4.83], $P = 0.003$) increased risk of asthma. The allelic test showed the A allele was associated with a 1.63-fold increased risk of asthma (95% CI [1.25–2.07], $P = 0.0004$). The AG plus AA genotypes were also associated with severe symptoms ($P = 0.0148$). Furthermore, the AG and AA genotype carriers had lower serum MEG3 expression level than the GG genotype carriers, consistent with the reported downregulation of MEG3 in asthma patients.

**Conclusion.** *MEG3* SNP rs7158663 is a genetic susceptibility locus for asthma in Taiwanese. Individuals carrying the variant genotypes have lower serum MEG3 level and are at increased risks of asthma and severe symptoms.

## INTRODUCTION

Asthma is the most common chronic lung disease characterized by airway obstruction, airway inflammation, and airway hyperresponsiveness. Globally, approximately 300 million people are affected by asthma annually, and its prevalence is continuously increasing (*Mattiuzzi & Lippi, 2020*; *for Asthma, 2022*). There are large geographical differences in the incidence of asthma and developed countries generally have higher incidences of asthma than developing countries largely due to higher environmental exposures including smog and air particles (*Mulgirigama et al., 2019*). Asthma is a complex disease whose etiology involves both environmental exposures and genetic susceptibility. It was estimated that asthma has a genetic heritability of up to 60%–80% (*Ober & Yao, 2011*; *Kabesch & Tost, 2020*; *Bonnelykke & Ober, 2016*). An animal disease model mimicking human asthma suggested that approximately 200 genes may contribute to the etiology of asthma (*Temesi et al., 2014*). Candidate gene studies focusing on biologically relevant genes such as inflammatory and immunological genes and genome-wide association studies (GWAS) have identified a number of genetic susceptibility loci for asthma (*Bonnelykke & Ober, 2016*; *Liu et al., 2022*; *Shen et al., 2017*; *Hsia et al., 2015*; *Garcia-Sanchez et al., 2015*; *Shi, Zhang & Qiu, 2022*; *Ranjbar et al., 2022*). However, the identified asthma susceptibility loci to date only explain a small portion of the genetic heritability of asthma and additional genetic susceptibility loci to asthma remains to be uncovered.

Long noncoding RNAs (lncRNAs) are defined as RNAs longer than 200 nucleotides that are not translated into functional proteins (*Orom & Shiekhattar, 2013*). In recent years, it has been increasingly evident that lncRNAs are important regulators of gene expression in diverse cellular processes, thus are essential for maintaining normal physiology, and are often dysregulated in various diseases including asthma (*Melissari & Grote, 2016*; *Rafiee et al., 2018*; *Statello et al., 2021*; *Gysens et al., 2022*; *Chen & Deng, 2022*). A number of studies have compared the expression of different lncRNAs in blood cells between asthma patients and healthy controls and identified several differentially expressed lncRNAs (*Tsitsiou et al., 2012*; *Booton & Lindsay, 2014*; *Feng, Yang & Yan, 2020*; *Chen & Deng, 2022*), among which maternally expressed gene 3 (MEG3) is of particular interest. The *MEG3* gene is located in chromosome 14q32.3 and encodes a lncRNA of approximately 1.6 kb (*Zhang et al., 2010*). MEG3 is abundantly expressed in normal tissues and plays an essential role in cell growth and organ development (*da Rocha et al., 2008*). Earlier studies found reduced expression of MEG3 in the circulating CD8+ T cells of patients with severe asthma (*Tsitsiou et al., 2012*; *Booton & Lindsay, 2014*). A recent study (*Feng, Yang & Yan, 2020*) reported that serum MEG3 level was significantly lower in asthma patients than in healthy controls and was the lowest in the most severe asthma patients. Furthermore, there was a significant inverse correlation between serum MEG3 expression and the course of asthma (r = $-0.666$, $P < 0.001$) (*Feng, Yang & Yan, 2020*). *In vitro* experiments also provide indirect evidence supporting that low MEG3 expression is linked to asthma development: treating human bronchial epithelial cells with cigarette smoke condensate (*Hu et al., 2009*) or an environmental carcinogen nickel (*Zhou et al., 2017*) caused marked downregulation of MEG3 expression.

Given that aberrant MEG3 expression is linked to asthma development, we hypothesize that genetic variants that affect MEG3 expression may be associated with the risk of developing asthma. Several SNPs in MEG3 have been reported in literature that may affect MEG3 expression and influence the risks of various diseases including inflammatory response, diabetes, stroke, and cancer (*Wallace et al., 2010*; *Han et al., 2018*; *Ghaedi et al., 2018*; *Ghafouri-Fard & Taheri, 2019*; *Gao et al., 2021*; *Zhu et al., 2021*; *Zhong et al., 2022*). We therefore perform the first study to evaluate the associations of SNPs in MEG3 with the risks of asthma using a case control study design. In addition, we determine whether the selected SNPs influence the expression level of MEG3 in serum samples and provide strong genotype-gene expression correlation that explains the observed significant association between SNPs and asthma risk.

## MATERIALS & METHODS

### Recruitment of asthmatic cases and non-asthmatic healthy controls

A total of 198 patients with asthma and 453 non-asthmatic healthy controls were recruited from China Medical University Hospital (CMUH) as previously described (*Hsia et al., 2015*; *Li et al., 2021*). The diagnosis of asthma was based on the following inclusion criteria: (1) more than two or three episodes of wheezing and shortness of breath during the past year; (2) diagnosis of asthma by pulmonologists together with the demonstration of reversible and variable airflow obstruction by spirometry; (3) symptoms; and (4) prescription of medications for asthma. No children were recruited (the youngest being 25 years old) and there was no sex restriction. The controls were selected by frequency-matching to cases by age and sex after initial random sampling from the Health Examination Cohort of CMUH. The inclusion criteria for controls were: (1) no past or present physician's diagnosis of asthma and other pulmonary diseases; (2) no history of wheezing, shortness of breath, or other symptoms of allergic diseases such as nasal and skin symptoms; (3) no use of medications for asthma; (4) absence of first-degree relatives with a history of asthma; and (5) older than 25 years. Those with chronic inflammatory responses, diabetes, stroke and cancer were also excluded for both cases and controls, the flow chart of participant recruitment scheme was shown in Fig. 1. The symptom severity for each asthma case was verified by two experienced pulmonologists according to the Global Initiative for Asthma (GINA) guidelines (*for Asthma, 2022*). Specifically, symptom severities were classified into four groups based on the treatment to control the symptoms and exacerbations. Group 1 (mildest): treated with as-needed inhaled corticosteroid (ICS)-formoterol alone; Group 2, treated with low-intensity maintenance controller treatment of ICS-formoterol, leukotriene receptor antagonists or chromones; Group 3, treated with low dose ICS-long acting $\beta2$ agonist (LABA); and Group 4, treated (severest) with high dose ICS-LABA. Peripheral blood was collected from all subjects and genomic DNA was extracted and stored until genotyping (*Yang et al., 2017b*). This study was approved by the Research Ethics Committee of the China Medical University Hospital (CMUH106-REC1-004). All protocols were conducted in accordance with relevant guidelines. All patients provided written informed consent at the time of recruitment.
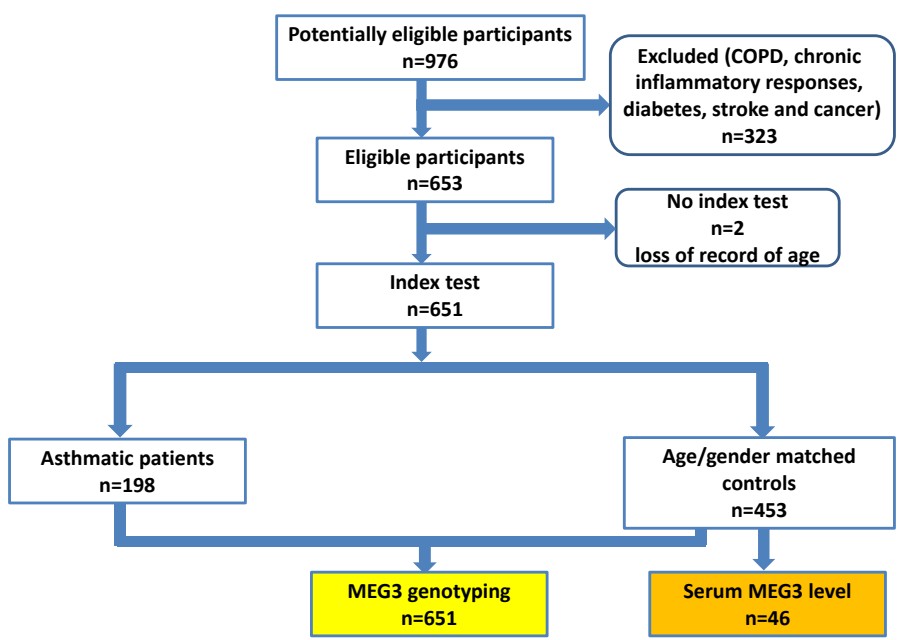

**Figure 1** The proposed flow chart of participant recruitment.

## Genotyping methodology for *MEG3* genotypes

We selected four SNPs (Fig. 2) that were either correlated with MEG3 expression or associated with other diseases in previous publications (*Han et al., 2018*; *Ghaedi et al., 2018*; *Ghafouri-Fard & Taheri, 2019*; *Gao et al., 2021*; *Zhu et al., 2021*; *Zhong et al., 2022*). The *MEG3* rs7158663, rs3087918, rs11160608, and rs4081134 genotypes were determined using TaqMan assay with an ABI 7500 Real-Time PCR System (Applied Biosystems, Foster City, CA, USA) as previously described (*Pei et al., 2022*).

## Transcriptional expression of MEG3 in serum

To evaluate the correlation between MEG3 RNA expression and *MEG3* rs7158663 genotype, 46 serum samples were randomly selected from the controls and subjected to extraction of total RNA using TRIzol Reagent (Invitrogen, Carlsbad, CA, USA). They were randomly selected from the pool of controls, and there were no significant differences between these samples and the remaining controls in terms of age, sex and smoking status. The expression levels of MEG3 were measured by real-time quantitative reverse transcription-PCR (RT-PCR) using an FTC-3000 real-time PCR instrument series (Funglyn Biotech Inc., Canada). GAPDH was used as an internal control (*Liao et al., 2020*; *Huang et al., 2018*). MEG3 primers were purchased from Qiagen (UniGene No. Hs. 654863, Catalog No. 4331182, LPH02974A-200). For GAPDH, the forward and reverse primers were 5′-GAAATCCCATCACCATCTTCCAGG-3′and 5′-GAGCCCCAGCCTTCTCCATG-3′, respectively. During the analysis, the levels of MEG3 RNA were normalized to those of GAPDH expression. Each sample was measured three times.

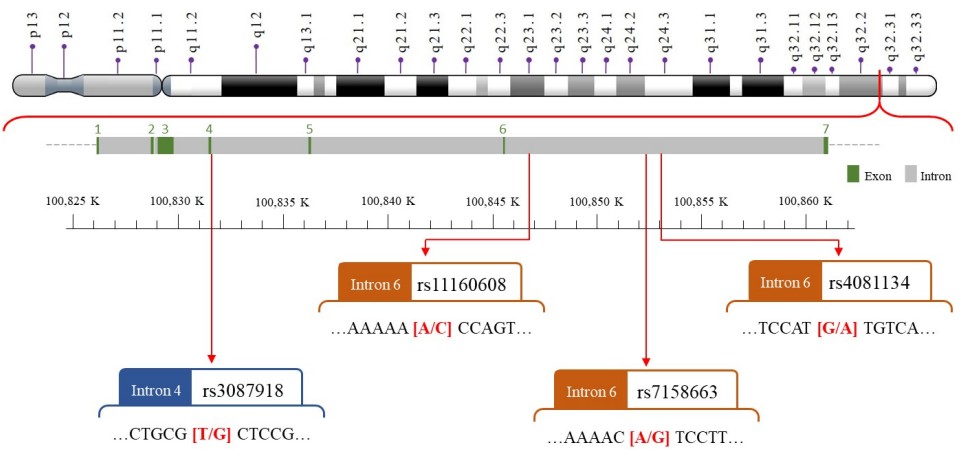

**Figure 2  The map of polymorphic sites of *MEG* rs7158663, rs3087918, rs11160608, and rs4081134.**

## Statistical methodology

The goodness-of-fit chi-square test was used to examine the fitness of the Hardy-Weinberg equilibrium in the control group. The Student's *t*-test was used to examine the differential distributions of age between the case and control groups. Pearson's chi-squared test was used to examine the differential distribution of various *MEG3* genotypes and the interaction between *MEG3* genotypes and symptom severity. The association of *MEG3* genotypes with asthma risk was evaluated by calculating the odds ratios (ORs) and the corresponding 95% confidence intervals (CIs) using multivariable logistic regression analyses adjusting for age, sex, and smoking status. The differential expression levels of MEG3 shown in Fig. 3 were evaluated with unpaired Student's *t*-test. The statistical significance level was set at $P < 0.05$.

## RESULTS

### Demographics of asthmatic and non-asthmatic subjects

The age, sex, smoking, and clinical characteristics such as pulmonary function and symptom severity of the 198 patients and 453 controls are shown in Table 1. The patients and controls were frequency-matched on age and sex ($P = 0.2972$ and 0.9956, respectively). There were slightly more ever-smokers among cases than controls (29.8% *vs.* 28%), but the difference was not statistically significant ($P = 0.7161$). Regarding pulmonary function, the average ratio of forced expiratory volume in the first second to forced vital capacity (FEV1/FVC, %) and the percentage of predicted FEV1 (FEV1%) was significantly lower in the asthmatic group than in the control group (both $P < 0.0001$). The percentage of patients within symptom severity group 1 (mild), 2, 3, and 4 (severe) was 30.3, 32.8, 17.2, and 19.7, respectively (Table 1).

### Associations of *MEG3* genotypes with asthma risk

The distributions of the genotypes of *MEG3* rs7158663, rs3087918, rs11160608, and rs4081134 in the cases and controls are shown in Table 2. First, the genotype frequencies

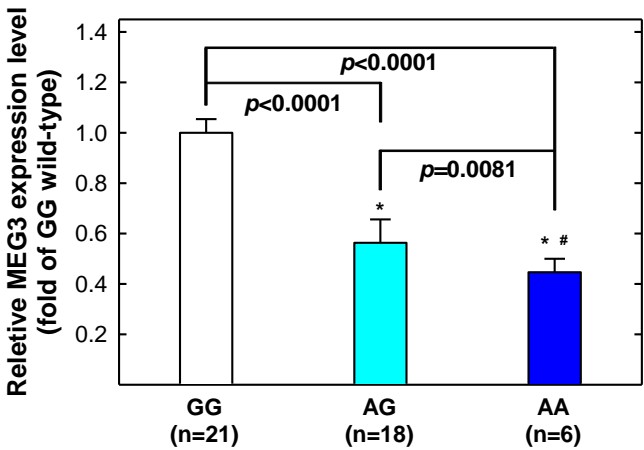

**Figure 3 Correlation between *MEG3* rs7158663 genotype and MEG3 expression in the serum of healthy controls.** An asterisk (*) indicates statistically significantly difference from GG genotype; # statistically significantly different from AG genotype.

**Table 1 Distributions of baseline characteristics among the 198 asthmatic patients and 453 controls.**

| Index | | Controls ($n = 453$) | | Cases ($n = 198$) | | P-value[a] |
|---|---|---|---|---|---|---|
| | | *n* | % | *n* | % | |
| Age (years) | 25–40 | 285 | 63.4% | 133 | 67.2% | |
| | >40 | 168 | 36.6% | 65 | 32.8% | 0.2972 |
| Gender | Male | 190 | 41.9% | 83 | 41.9% | |
| | Female | 263 | 58.1% | 115 | 58.1% | 0.9956 |
| Smoking status | Non-smoker | 326 | 72.0% | 139 | 70.2% | |
| | Smoker | 127 | 28.0% | 59 | 29.8% | 0.7161 |
| Pulmonary functions | (mean ± SD) | | | | | |
| FEV1/FVC (%) | | 80.8 ± 8.1 | | 62.0 ± 13.0 | | <0.0001 |
| FEV1% | | 92.9 ± 5.8 | | 69.1 ± 12.9 | | <0.0001 |
| Symptoms severity | | | | | | |
| 1 (mild) | | | | 60 | 30.3% | |
| 2 | | | | 65 | 32.8% | |
| 3 | | | | 34 | 17.2% | |
| 4 (severe) | | | | 39 | 19.7% | |

**Notes.**

Abbreviation: FEV1, forced expiratory volume in first second; FVC, forced vital capacity; FEV1%, percent of predicted FEV1.

[a] Chi-square without Yate's correction test or Student's *t*-test.

of the four SNPs in the control group all fit well with Hardy-Weinberg equilibrium (all $P > 0.05$). Second, the genotype frequencies of rs7158663 were differentially distributed between the cases and controls with the heterozygous variant AG and homozygous variant AA genotypes over-represented in the cases compared to controls (44.5% *vs.* 36.4% for AG and 12.1% *vs.* 6.6% for AA genotype). In logistic regression analyses, the AG and AA genotypes were both associated with increased risks of asthma (adjusted ORs = 1.62

**Table 2** Distributions of *MEG* various genotypes among asthmatic patients and non-asthmatic controls.

| Genotype | Asthmatic cases, *n* (%) | Non-asthmatic controls, *n* (%) | Adjusted OR (95% CI)[a] | *P*-value[b] |
|---|---|---|---|---|
| rs7158663 | | | | |
| GG | 86 (43.4) | 258 (57.0) | 1.00 (Reference) | |
| AG | 88 (44.5) | 165 (36.4) | **1.62 (1.18-2.32)** | **0.0093**[*] |
| AA | 24 (12.1) | 30 (6.6) | **2.68 (1.52-4.83)** | **0.0030**[*] |
| AG+AA | 112 (56.6) | 195 (43.0) | **1.75 (1.29-2.48)** | **0.0015**[*] |
| $P_{trend}$ | | | | **0.0024**[*] |
| $P_{HWE}$ | | | | 0.6039 |
| rs3087918 | | | | |
| TT | 74 (37.4) | 183 (40.4) | 1.00 (Reference) | |
| GT | 94 (47.5) | 215 (47.5) | 1.12 (0.79–1.65) | 0.6732 |
| GG | 30 (15.1) | 55 (12.1) | 1.37 (0.85–2.31) | 0.2588 |
| GT+GG | 124 (62.6) | 240 (59.6) | 1.29 (0.87–1.81) | 0.1650 |
| $P_{trend}$ | | | | 0.5285 |
| $P_{HWE}$ | | | | 0.5014 |
| rs11160608 | | | | |
| AA | 65 (32.8) | 136 (30.0) | 1.00 (Reference) | |
| AC | 98 (49.5) | 232 (51.2) | 0.85 (0.62–1.31) | 0.5221 |
| CC | 35 (17.7) | 85 (18.8) | 0.87 (0.55–1.43) | 0.5527 |
| AC+CC | 133 (67.2) | 317 (70.0) | 0.89 (0.62–1.28) | 0.4758 |
| $P_{trend}$ | | | | 0.7710 |
| $P_{HWE}$ | | | | 0.4256 |
| rs4081134 | | | | |
| GG | 104 (52.5) | 254 (56.0) | 1.00 (Reference) | |
| AG | 82 (41.4) | 172 (38.0) | 1.18 (0.84–1.68) | 0.3915 |
| AA | 12 (6.1) | 27 (6.0) | 1.07 (0.55–2.27) | 0.8226 |
| AG+AA | 94 (47.5) | 199 (44.0) | 1.11 (0.87–1.64) | 0.4029 |
| $P_{trend}$ | | | | 0.6920 |
| $P_{HWE}$ | | | | 0.7657 |

Notes.

OR, Odds ratio; CI, confidence interval; HWE, Hardy-Weinberg Equilibrium; $P_{trend}$, *P*-value for trend analysis; $P_{HWE}$, *P*-value for Hardy-Weinberg equilibrium analysis.

[a] Data has been adjusted for confounding factors for asthma including age, gender and smoking.

[b] Based on Chi-square test with Yates' correction.

[*] The significant *p*-value are in bold and marked with an asterisk.

and 2.68, 95% CIs [1.18−2.32] and [1.52−4.83], $P = 0.0093$ and 0.0030, respectively) compared with the wild-type GG genotype. In the dominant model, individuals carrying the variant genotypes (AG+AA) exhibited a 1.75-fold increased risk of asthma (adjusted OR = 1.75, 95% CI [1.29−2.48], $P = 0.0015$). The other three SNPs, rs3087918, rs11160608, and rs4081134, were not significantly associated with the risks of asthma (Table 2).

## Allelic frequency distribution analysis

The allelic frequencies of rs7158663, rs3087918, rs11160608, and rs4081134 SNPs among cases and controls are shown in Table 3. Consistent with the findings in Table 2, individuals

**Table 3  Distribution of *MEG* allelic frequencies among asthmatic patients and non-asthmatic controls.**

| Allelic type | Asthmatic cases, *n* (%) | Non-asthmatic controls, *n* (%) | Adjusted OR (95% CI)[a] | *P*- balue[b] |
|---|---|---|---|---|
| rs7158663 | | | | |
| Allele G | 260 (65.7) | 681 (75.2) | 1.00 (Reference) | |
| Allele A | 136 (34.3) | 225 (24.8) | **1.63 (1.25-2.07)** | **0.0004*** |
| rs3087918 | | | | |
| Allele T | 242 (61.1) | 581 (64.1) | 1.00 (Reference) | |
| Allele G | 154 (38.9) | 325 (35.9) | 1.14 (0.85–1.48) | 0.2990 |
| rs11160608 | | | | |
| Allele A | 228 (57.6) | 504 (55.6) | 1.00 (Reference) | |
| Allele C | 168 (42.4) | 402 (44.4) | 0.89 (0.74–1.19) | 0.5148 |
| rs4081134 | | | | |
| Allele G | 290 (73.2) | 680 (75.1) | 1.00 (Reference) | |
| Allele A | 106 (26.8) | 226 (24.9) | 1.08 (0.84–1.47) | 0.4875 |

**Notes.**

OR, Odds ratio; CI, confidence interval.

[a] Data has been adjusted for confounding factors for asthma including age, gender and smoking.

[b] Based on Chi-square test with Yates' correction.

*The significant *p*-value are in bold and marked with an asterisk.

with the variant A allele at rs7158663 were at a higher risk of asthma than those with the wild-type G allele after adjusting for age, sex, and smoking status (adjusted OR = 1.63, 95% CI [1.25−2.07], $P = 0.0004$). In contrast, the variant alleles of the other three SNPs were not associated with asthma risks (Table 3).

### *MEG3* rs7158663 genotypes are associated with the severity of asthma symptom

We then analyzed the distributions of *MEG3* genotypes among asthma patients within different symptom groups (Table 4). The variants genotypes (AG or AA) of rs7158663 were over-represented in patients with more severe symptoms: the frequencies of the variant genotypes and wild-type (GG) genotype was 35% and 65% in patients with the mildest symptom (Group 1), 44.6% and 55.4% in Group 2, 55.9% and 44.1% in Group 3, and 69.2% and 30.8%, respectively, in patients with the most severe symptom (Group 4) ($P = 0.0148$) (Table 4). In contrast, the other three SNPs were not associated with asthma symptoms.

To determine whether the association of rs7158663 with asthma differs between patients with mild and severe symptoms, we next performed stratified analyses. To increase statistical power, we combined the two milder symptom groups and two severer symptom groups. As shown in Table 5, the association of the rs7158664 variant genotypes with asthma risk was only significant in patients with severer symptoms (adjusted OR = 2.48, 95% CI [1.44−4.26], $P = 0.0024$), but not in patients with milder symptoms (adjusted OR = 0.91, 95% CI [0.61−1.43], $P = 0.6115$) (Table 5).

**Table 4  Association of *MEG* polymorphisms with the symptoms severity among asthmatic patients.**

| Genotype | Symptom severity, n (%) | | | | P- value[a] |
|---|---|---|---|---|---|
| | 1 (mild) | 2 | 3 | 4 (severe) | |
| rs7158663 | | | | | |
|   Wild-type genotype | 39 (65.0) | 36 (55.4) | 15 (44.1) | 12 (30.8) | |
|   Variant genotypes | 21 (35.0) | 29 (44.6) | 19 (55.9) | 27 (69.2) | **0.0148**[*] |
| rs3087918 | | | | | |
|   Wild-type genotype | 33 (55.0) | 31 (47.7) | 17 (50.0) | 18 (46.2) | |
|   Variant genotypes | 27 (45.0) | 34 (52.3) | 17 (50.0) | 21 (53.8) | 0.8941 |
| rs11160608 | | | | | |
|   Wild-type genotype | 31 (51.7) | 33 (50.8) | 16 (47.1) | 17 (43.6) | |
|   Variant genotypes | 29 (48.3) | 32 (49.2) | 18 (52.9) | 22 (56.4) | 0.9452 |
| rs4081134 | | | | | |
|   Wild-type genotype | 34 (56.7) | 32 (49.2) | 17 (50.0) | 18 (46.2) | |
|   Variant genotypes | 26 (43.3) | 33 (50.8) | 17 (50.0) | 21 (53.8) | 0.8526 |

**Notes.**
[a] Chi-square without Yate's correction test.
[*] The significant *p*-value are in bold and marked with an asterisk.

**Table 5  Association of *MEG* rs7158663 genotypes with the symptoms severity among stratified asthmatic patients.**

| Genotype | Asthmatic cases, *n* (%) | Non-asthmatic controls, *n* (%) | Adjusted OR (95% CI)[a] | P-value[b] |
|---|---|---|---|---|
| Milder symptom severity | | | | |
| Wild-type genotype | 75 (60.0) | 258 (57.0) | 1.00 (Ref) | |
| Variant genotypes | 50 (40.0) | 195 (43.0) | 0.91 (0.61–1.43) | 0.6115 |
| Severer symptom severity | | | | |
| Wild-type genotype | 27 (37.0) | 258 (57.0) | 1.00 (Ref) | |
| Variant genotypes | 46 (63.0) | 195 (43.0) | **2.48 (1.44–4.26)** | **0.0024**[*] |

**Notes.**
OR, Odds ratio; CI, confidence interval.
[a] Data has been adjusted for confounding factors for asthma including age, gender and smoking.
[b] Based on Chi-square test with Yates' correction.
[*] The significant *p*-value are in bold and marked with an asterisk.

## The variant genotypes of rs7158663 correlated with lower serum MEG3 level

Since *MEG3* rs7158663 variant genotypes are associated with increased asthma risks, and asthma patients have lower serum MEG3 level than controls (*Feng, Yang & Yan, 2020*), we next determine whether there is a correlation between rs7158663 genotypes and serum MEG3 level. We randomly selected 46 healthy controls and measured serum MEG3 expression level by quantitative PCR (one sample was somehow technically undetectable). Compared to the wild-type GG genotype carriers, the serum MEG3 expression level was significantly lower for the GA genotype carriers ($P < 0.0001$) and the lowest in AA genotype carriers ($P < 0.0001$ *vs.* GG genotype carriers and $P = 0.0081$ *vs.* AG genotype carriers) (Fig. 3).

## DISCUSSION

In the current study, we found significant associations between *MEG3* rs7158663 SNP and the risk of asthma and symptom severity. Specifically, the variant genotypes (AG and AA) of rs7158663 were associated with increased risks of asthma and severe asthma symptoms. The variant genotype carriers had significantly lower serum MEG3 expression (Fig. 3), consistent with prior reports of downregulation of MEG3 expression in asthma patients and providing biological explanation for the observed associations between variant genotypes and increased asthma risks. To our knowledge, the present study is the first to report the associations of *MEG3* genotypes with asthma risk and symptom severity.

MEG3 is a lncRNA that is abundantly expressed in most normal tissues but downregulated in a variety of tumor tissues including lung cancer (*Ghafouri-Fard & Taheri, 2019*). There have been strong evidences supporting that MEG3 plays a role in asthma development. An early study compared the profile of lncRNA expression in circulating $CD8^+$ T cells from asthma patients and healthy controls and found MEG3 expression was significantly lower in patients with severe asthma than healthy controls (*Tsitsiou et al., 2012*). A recent study (*Feng, Yang & Yan, 2020*) compared serum MEG3 expression level in 119 asthma patients and 125 healthy controls and found significantly lower serum MEG3 level in asthma patients than in controls and the expression level was the lowest in mixed granulocytic asthma (the subtype with the most severe symptoms). Furthermore, serum MEG3 level was negatively correlated with the course of disease (r = −0.666, $P < 0.001$). Logistic regression analysis showed that inflammatory phenotype, the course of disease, and serum MEG3 level were independent prognostic factors for the recurrence of asthma (*Feng, Yang & Yan, 2020*). Supporting these direct evidences of human studies linking low MEG3 expression to asthma development, an early *in vitro* experiment tested the effects of cigarette smoke condensate (CSC) treatment on gene expression of human bronchial epithelial cells (HBEC) and found MEG3 was down-regulated in CSC-treated cells (*Hu et al., 2009*). Another *in vitro* study showed MEG3 expression was suppressed upon nickel (an environmental toxigenic molecule) treatment of HBEC (*Zhou et al., 2017*). All these evidences suggest that abundant MEG3 expression is important to maintain normal lung function and physiology, whereas reduced MEG3 expression favors proliferation and promotes the pathogenesis of common lung diseases such as asthma and lung cancer.

Our findings in this current study supports the notion that reduced MEG3 expression favors asthma development. We found the variant genotypes (AG and AA) of *MEG3* rs7158663 were associated with increased risks of asthma and severe symptoms. Furthermore, the variant genotypes of rs7158663 correlated with lower serum MEG3 expression. The rs7158663 genotypes-MEG3 expression-asthma relationship is therefore biologically meaningful and plausible. Consistent with our rs7158663 genotype-MEG3 expression correlation, a previous study also found the variant A allele of rs7158663 was associated with significantly decreased serum MEG3 level compared to the wild type G allele ($P < 0.0001$) (*Ali et al., 2020*).

The exact biological mechanisms underlying the roles of MEG3 in asthma pathogenesis remains unclear. MEG3 is a multi-faceted molecule involved in many microRNA and

protein interactions, signal pathways, and cellular processes (*Ghafouri-Fard & Taheri, 2019*; *Chen & Deng, 2022*). MEG3 activates transcription factor and tumor suppressor TP53 by stabilizing TP53 protein and/or augmenting its transcriptional activity, thereby induces TP53-dependent target gene expression and inhibits cell growth (*Zhou et al., 2007*; *Ghafouri-Fard & Taheri, 2019*). MEG3 interacts with the polycomb repressive complex 2 (PRC2), which catalyzes the methylation of histone H3 lysine 27 (H3K27) and functions as a key epigenetic regulator for normal development (*Simon & Kingston, 2013*). MEG3 interacts with PRC2 and its cofactor JARID2 to regulate TGF-$\beta$ signaling pathway genes (*Kaneko et al., 2014*; *Mondal et al., 2015*). MEG3 also contributes to the regulation of several other important cellular signaling pathways, including PI3K/AKT/mTOR, Wnt/$\beta$-catenin, JAK/STAT, and Notch, and these pathways have been implicated in asthma (*Wang et al., 2014*; *Ghafouri-Fard & Taheri, 2019*; *Athari, 2019*; *Zhao et al., 2020*. More recently, increasing studies have shown another important function of MEG3: serving as a competing endogenous RNA (ceRNA) and sponge for various miRNAs, and counteracting their regulatory effects on target genes (*Moradi, Fallahi & Rahimi, 2019*; *Li et al., 2016*; *Zhang et al., 2021*; *Li et al., 2018*; *Luo et al., 2022*; *Qiu et al., 2019*; *Dong et al., 2021*; *Sun et al., 2021*; *Wang et al., 2021*). In this regard, it is worth noting that MEG3 could sponge microRNA-125a-5p (*Li et al., 2016*) and miRNA-17 (*Qiu et al., 2019*) as a ceRNA, relieve their inhibition on orphan receptor $\gamma$t (ROR $\gamma$t), which is a regulator of regulatory T (Treg) cells and helper T cell-17 (Th17). Mounting evidences suggest that the balance of Treg and Th17 (Treg/Th17) plays a critical role in asthma development (*Tao et al., 2015*; *Zhu et al., 2017*; *Jing, Wang & Liu, 2019*; *Zhou et al., 2022*). Therefore, the microRNA sponge effect of MEG3 may be one of the biological mechanisms responsible for the involvement of MEG3 in asthma development. Better understanding of the biology behind the roles of MEG3 in asthma warrants further investigation.

MEG3 SNPs are associated with multiple diseases including asthma, inflammatory response, diabetes, stroke, and cancer (*Wallace et al., 2010*; *Han et al., 2018*; *Ghaedi et al., 2018*; *Ghafouri-Fard & Taheri, 2019*; *Gao et al., 2021*; *Zhu et al., 2021*; *Zhong et al., 2022*). Pleiotropy (*i.e.,* shared genetic predisposition loci to different human diseases and traits) is pervasive in human genome (*Sivakumaran et al., 2011*; *Pickrell et al., 2016*; *Tian et al., 2022*). For example, a missense SNP (rs13107325) the zinc transporter SLC39A8 influences the risks of at least seven different diseases and genetic traits, including schizophrenia, Parkinson's disease, Crohn's disease, allergy, height, nearsightedness, and HDL (*Pickrell et al., 2016*); and SNPs in the TERT-CLPTM1L locus have been associated with more than 12 different cancer and non-cancer diseases (*Tian et al., 2022*).

The present study has several limitations. First, we only selected one candidate gene and the sample size was modest. Larger scale studies are needed to have a more complete understanding of the genetic susceptibility to asthma in Taiwan. Second, we had missing or incomplete information on some known environmental risk factors and lacked longitudinal data of the dynamic environmental exposures and could not perform gene-environment interaction analyses. Lastly, the mRNA levels of MEG3 are lower in asthma patients than in healthy controls (*Feng, Yang & Yan, 2020*). As a common practice, we performed the

genotype-expression correlation in a subset of healthy controls. It is worthwhile to validate that the variant genotypes also correlate with lower MEG expression in asthma patients.

## CONCLUSIONS

This study provides the first evidence that the variant genotypes at *MEG3* rs7158663 are associated with increased asthma risk and symptom severity. Moreover, there is a significant association between the variant genotype and lower MEG3 expression in serum. *MEG3* rs7158663 is a novel genetic susceptibility locus for asthma.

## ACKNOWLEDGEMENTS

We thank Yu-Ting Chin and Tai-Lin Huang for technical assistance in DNA extraction and serum MEG3 measurement.

### Funding

This study was supported by grants from Taichung Tzu Chi Hospital (TTCRD111-15), China Medical University Hospital, and Asia University (CMU111-ASIA-02). The funders had no role in study design, data collection and analysis, decision to publish, or preparation of the manuscript.

### Grant Disclosures

The following grant information was disclosed by the authors:
Taichung Tzu Chi Hospital: TTCRD111-15.
China Medical University Hospital.
Asia University: CMU111-ASIA-02.

### Competing Interests

The authors declare there are no competing interests.

### Author Contributions

- Kuo-Liang Chiu conceived and designed the experiments, analyzed the data, authored or reviewed drafts of the article, and approved the final draft.
- Wen-Shin Chang performed the experiments, prepared figures and/or tables, authored or reviewed drafts of the article, and approved the final draft.
- Chia-Wen Tsai performed the experiments, analyzed the data, prepared figures and/or tables, authored or reviewed drafts of the article, and approved the final draft.
- Mei-Chin Mong conceived and designed the experiments, authored or reviewed drafts of the article, and approved the final draft.
- Te-Chun Hsia analyzed the data, authored or reviewed drafts of the article, and approved the final draft.
- Da-Tian Bau analyzed the data, authored or reviewed drafts of the article, and approved the final draft.

## Human Ethics

The following information was supplied relating to ethical approvals (i.e., approving body and any reference numbers):

The study was approved by the Ethics Review Committee of the China Medical University Hospital (CMUH106-REC1-004).

## Data Availability

The raw measurements are available in the Supplementary Files.

## Supplemental Information

Supplemental information for this article can be found online at http://dx.doi.org/10.7717/peerj.14760#supplemental-information.

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
