# Peer review of "Novel genetic variants in long non-coding RNA MEG3 are associated with the risk of asthma"

_PeerJ, doi:10.7717/peerj.14760_

## Round 0.1 · original submission · Major Revisions

According to the reviewers' comments, I recommend reading carefully and attending it.

Reviewer 1 ·

Basic reporting

Abstract
- The present study proposes that Novel genetic variants in long non-coding RNA MEG3 are associated with the risk of asthma However several questions can be addressed:
In the Background: the authors mention “Asthma is chronic airway inflammation, and its diagnosis and treatment strategies remain challenging.” However, throughout this study, statistical tools are not used to evaluate the long non-coding RNA MEG3 or its genotypes as a possible marker or diagnostic tool, and much less is its use assessed as a treatment for asthma. it is suggested to modify the wording.
- In lines 34-46 it is mentioned “No study has confirmed the contribution of MEG3, a tumor suppressor, to asthma.” Therefore, the importance of long non-coding RNA MEG3 in asthma is not explained or why it is essential for this study, so it is suggested to finish explaining the idea or to give more detail to this part.
- In lines 51-52 the authors mentioned that “MEG3 as a novel signature for early asthma detection but also for prognosis prediction.” However, in the present study, no statistical tools are presented to evaluate MEG3 as a diagnosis of asthma, much less an early diagnosis. It is suggested to modify the text.
- In line 52 they mention “detection but also for prognosis prediction.” In this study, no statistical tools are presented to support this statement. The present study does not have the necessary information or the follow-up of the patients to calculate the function of MEG3 as a prognostic tool. So it is suggested to modify the writing.
Introduction
- The introduction goes too far into the topic of MEG3 in cancer, however, they do not mention a sufficiently clear relation with ASMA.
- In line 88 it is mentioned that “MEG3 genotypes on asthma risk has never been investigated”. So where or how the idea of investigating it comes from, it is not explained where this idea comes from, in general, I consider that it is a repetitive theme throughout the present study, the relation between MEG3 and ASMA is not clear.
- Something that I consider extremely important is that the authors do not present previous tools to determine the relationship of the long non-coding RNA MEG3 in asthma, and yet the authors delve into the association of the MEG3 SNPs, without any previous clare relationship between MEG3-ASMA.

Experimental design

Materials and methods.
- It is not specified whether the control cases had any lung-related disease. This is because not even the authors themselves fully understand the pathway and the target genes or target proteins of MEG3, which could be related to other lung diseases and not only to ASMA.
- Was the ASTHMA severity classification also based on GINA 2022? It is not clear, it is recommended to add this information.
- In the section “Genotyping methodology for MEG3 genotypes”, the authors do not mention how and why these MEG3 genotypes were selected, in addition to their importance.

Validity of the findings

Results
- The authors mention in lines 149-152 that “Regarding pulmonary function, the average ratio of forced expiratory volume in the first second to forced vital capacity (FEV1/FVC, %) and the percentage of predicted FEV1 (FEV1%) was lower in the asthmatic group than in the control group (both P < 0.0001).” However, it does not provide the necessary information to identify whether patients may have underlying lung disease, because if the pathway and the target genes or target proteins of MEG3 are not fully understood, it could be related to other lung diseases and not only an ASTHMA, it is suggested to review.
- It is suggested addition to the tables or the description of the patients the information of the patients who were smokers because this variable acquires weight when calculating the OR.
- In lines 193-194 the authors mention “MEG3 rs7158663 were at a higher risk of severe symptom severity than wild-type”. Again, there are no tools that show risk of severe disease, that is, the study only refers to the severity of symptoms, not the risk of developing them.

- In line 202 and the materials and methods section, it is mentioned that 46 non-asthmatic individuals were selected to investigate the level of MEG3 expression, however, it is confusing and they do not mention how or why these 46 patients were selected.
- In the section “Transcriptional expression level of MEG3 among people“ they mention that “there was also a significant difference between the expression levels of MEG3 in the MEG3 rs7158663 AG genotype carriers and the 6 AA genotype carriers (P-values = 0.0081).” however, it is not clear what this result means, since the authors do not mention the relationship of MEG3 in healthy people, that is, there is no clear relationship between MEG3-ASMA or MEG3 and healthy people.

Discussion.
-Once again, the authors give too much weight to Cancer and the MEG3-ASMA relationship is poorly developed. It is suggested to retake the main objective of the study and reduce the content that alludes to Cancer.
-In lines 225-227, the authors keep repeating that “On the contrary, whether genetic variants of MEG3 contribute to the determination of the personal risk of asthma has never been examined.” So their results, how they are interpreted, or how important it is to solve this problem, furthermore, continue to show that the authors do not know why it would be essential to study MEG3 in ASTHMA.
- In lines 237-240 “MEG3 rs7158663 genotypes to symptom severity and found that variant genotype carriers of MEG3 rs7158663 were at a higher risk of suffering from severe symptoms than wild-type carriers.”, Again, this study lacks statistical tools to support this assertion. Risk is not calculated, only about severity. In addition, if they wanted to relate it to risk, variables that would be important, such as previous and current control of patients, comorbidities, exacerbations, etc., are not taken into account. Multiple factors could cause the patient to be in a more severe phase and not be related to MEG3, therefore, the risk is not assessed but only associated with severity. It is suggested to modify the text or add statistical tools to calculate the risk.
- In lines 242-244 “Our studies demonstrated that not only MEG3 rs7158663 genotypes can serve as a novel predictor for asthma risk but also as a signature for asthma patients suffering from severe symptoms”. Again, it lacks the tools to postulate MEG3 as a predictor of risk. Only the authors related it to symptom severity, but not to disease risk or progression.
-In lines 244-247 “MEG3 genotype contributes to severe symptoms are unknown, it can do lots of help to predict the prognosis of patients with asthma for more precise therapy and medication.” The present study lacks tools to postulate MEG3 as a marker or prognostic tool since they are not followed up over time and other variables such as exacerbations of ASTHMA or treatments used are not assessed. Only the authors related it to the severity of symptoms, but not to progression or prognosis.
- Along the same lines, the authors mention that “MEG3 genotype contributes to severe symptoms are unknown, it can do lots of help to predict the prognosis of patients with asthma for more precise therapy and medication.” It is very risky to make this assumption based on the fact that you reaffirm that there are no previous MEG3-ASMA relationships and you did not look for them either. In addition, these patients are not followed up to see if the treatment or the exacerbations can make changes in the long non-coding RNA MEG3.
- In lines 253-256 the authors mention a possible relationship of MEG3 with PI3K and AKT, however, there is no bibliographic evidence to support this statement. In general, the authors give a lot of weight to PI3K and AKT but again related to Cancer, it is suggested not to lose the main idea of MEG3-ASMA.
- In line 265 “The predictive value of MEG3 for asthma is remarkable in clinical practice.” But it is confusing, where does this statement come from if you state that there is no information on these two variables (MEG3-ASMA) furthermore, you did not investigate this relationship either.
In lines 312-313 “Thus, MEG3 may serve as an auxiliary diagnostic indicator of asthma and can be used for early screening in clinical practice.” Again, the present study lacks the tools to propose it or to know if it could work as a diagnostic marker or early diagnosis. Please consider changing the wording.
- Lines 69-271, “The prevalence of asthma is extremely high in cities, and there are many childhood or even baby asthmatic cases that make it difficult to go to hospitals” If the present study does not include children and they do not know how MEG3 would behave in this population, why mention them?

- 271-276 “The detection of MEG3 in serum is potentially much more convenient as long as the collection of peripheral blood and the preservation of RNA are countable and properly conducted. It is more convenient, less invasive, more objective (needs no subjective judgment or annotation), and can be conducive at any place, even outside of the hospital, when people are suffering from epidemic bursts of diseases, such as covid-19.” Is it more convenient, less invasive, and more objective than current methods? Clinically, I disagree with this assertion. Furthermore, even if it were so, not all hospitals have the necessary equipment or the necessary supplies to make these determinations.
- 280-282, “the environmental factors mentioned in the introduction may all contribute to asthma etiology; however, the current study had incomplete information about these factors, and some factors were still unknown.” Another point for which they lack tools to propose themselves as diagnostic tools.

Additional comments

The authors have incredible and important results, however, they are not exploited in the right way, leaving out important results. The present study lacks tools to support the conclusion since they do not present tools to propose it as a predictor of the appearance, diagnosis, or prognosis of ASTHMA. Consider rewriting them with objective results obtained in this study.

Reviewer 2 ·

Basic reporting

The authors present a manuscript a “Novel genetic variants in long non-coding RNA MEG3 are associated with the risk of asthma “


The authors demonstrated distribution of MEG3 rs7158663 genotypes was significantly different between the asthmatic and non-asthmatic control groups (P = 0.0024). In detail, variant AG and AA genotypes were of higher proportions in asthmatic group than in non-asthmatic group (OR = 1.62 and 2.68, 95% CI = 1.18-2.32 and 1.52-4.83).

There are several grammatical issues in this manuscript.

IN figure 2, Correlation between MEG3 rs7158663 genotype and MEG3 expression in the blood of non-asthmatic healthy subjects. The authors are indicating a relative expression of the GC, AG, or AA phenotype. Unfortunately, this is genoptyping data, and therefore language should be restricted to the presence of the MEG3 polymorphism or variant.
Also there were A total of 198 patients with asthma recruited in the study. Only 46 genotyping samples are displayed.
The authors should include a PRISMA flow chart or some other diagram indicating inclusion/exclusion criteria. 

The authors indicate rs7158663 Variant is linked with slightly more severity in asthma than other variants. P 0.0148. Upon looking the values, the difference is only 10 or so patients in one severity group over another.
The suggestion is to stratify into two severity groups and determine if this pattern is true (with larger numbers in you sampling population()

Experimental design

The authors should include a PRISMA flow chart or some other diagram indicating inclusion/exclusion criteria.

Validity of the findings

The authors indicate rs7158663 Variant is linked with slightly more severity in asthma than other variants. P 0.0148. Upon looking the values, the difference is only 10 or so patients in one severity group over another.
The suggestion is to stratify into two severity groups and determine if this pattern is true (with larger numbers in you sampling population()

---

## Round 0.2 · Minor Revisions

The reviewers have commented that most of the previous concerns have been attended to; thanks for that.

Now, some minor issues should be solved. I'll appreciate you can do it promptly.

Reviewer 1 ·

Basic reporting

I thank the authors for the response and the changes in the current work. It's evident that you worked hard and made a lot of changes. The work now is more precise and you focused on the MEG3-Asthma and not on cancer.

Now I identified a couple of details.

1. You say in line 110 "... literature that may affect MEG3 expression and influence the risk of various diseases including inflammatory response, diabetes, stroke and cancer." The ask is, does the study population have any of these diseases? or did you make any adjustments in the calculations to know that these variables do not interfere with the results? I suggest that you mention the answer in the corresponding area.

2. Is not clear yet how is a connexion between these SNVs and Asthma, looks like they have more connexions with other diseases than Asthma. I suggest adding a little description in the corresponding area, why these SNVs in Asthma?

3. In conclusion you say ..."Is novel genetic susceptibility locus for asthma in Taiwanese". I suggest modifying this sentence. although the study population is Taiwanese, this paper will be read around the world and could reduce the impact of your results on other populations.

Again I thank you for the changes in the whole text.

Experimental design

I reviewed the changes and I am satisfied with them.

Validity of the findings

I reviewed the changes and I am satisfied with them.

Additional comments

I reviewed the changes and I am satisfied with them.

Reviewer 2 ·

Basic reporting

Remaining comments -

Also there were A total of 198 patients with asthma recruited in the study. Only 46 genotyping samples are displayed.

Reviewer Response - A total of 198 healthy controls is required.

The authors should include a PRISMA flow chart or some other diagram indicating inclusion/exclusion criteria. 


Reviewer Response - inclusion/exclusion criteria is still required as a flow chart.

Experimental design

n/a

Validity of the findings

n/a

Additional comments

n/a

---

## Round 0.3 · accepted · Accept

Thank you for the new version of your manuscript.